# Transcriptome Sequencing and Differential Expression Analysis Reveal Molecular Mechanisms for Starch Accumulation in Chestnut

**Shengxing Li [1], Haiying Liang [2], Liang Tao [1], Liquan Xiong [1], Wenhui Liang [3], Zhuogong Shi [1,*] and Zhiheng Zhao [3,*]**

1   Key Laboratory for Forest Resources Conservation and Utilization in the Southwest Mountains of China, Ministry of Education, Southwest forestry University, Kunming 650224, China; shengxing_555@126.com (S.L.); basanyeyu@vip.sina.com (L.T.); liquanxiong@126.com (L.X.)
2   Department of Genetics and Biochemistry, Clemson University, Clemson, SC 29634, USA; hliang5499@gmail.com
3   Key Laboratory of Cultivation and Utilization of Guangxi Characteristic Economic Forest, Guangxi Forestry Research Institute, Nanning 530225, China; I.wenhui@163.com
*   Correspondence: zgongshi@sina.com (Z.S.); zzhhly09@163.com (Z.Z.); Tel.: +86-0771-2319955 (Z.Z.)

**Abstract:** Chestnuts are popular edible nuts that are rich in starch. In order to enhance the transcriptomic resources and further understand starch and sucrose metabolism in maturing chestnuts, a comparative transcriptomic study of Chinese chestnut kernels was conducted at three ripening stages (70, 82, and 94 DAF). At 82 and 94 days after flowering (DAF), starch continued to accumulate, and the amylopectin/amylose ratio increased. Transcriptomic profiling of kernels at 70 (stage I), 82 (stage II), and 94 DAF (stage III) indicated that soluble starch synthase and α-1,4-glucan branching enzyme genes are actively expressed at 82 and 94 DAF. The starch degradation enzymes amylase, phosphoglucan phosphatase DSP4, and maltose exporter did not show differential gene expression, while glycogen phosphorylase-encoding unigenes were significantly down-regulated at 94 DAF. In addition to starch and sucrose metabolism, RNA transport, RNA degradation, pyrimidine metabolism, purine metabolism, plant hormone signal transduction, plant–pathogen interactions, and glycerophospholipid metabolism were found to be significantly enriched in all comparisons included in the study. As Chinese chestnut matured, the unique enriched pathways switched from ribosomal biogenesis and RNA polymerase of eukaryotes to endocytosis and spliceosomes. These genomic resources and findings are valuable for further understanding starch and sucrose metabolism in the Chinese chestnut.

**Keywords:** amylopectin/amylose ratio; chestnut; gene differential expression; RNA sequencing; starch synthesis

## 1. Introduction

Chinese chestnut (*Castanea mollissima* Blume), a member of the Fagaceae family, is native to the Chinese mainland, Taiwan, and Korea. The species is well known for its nutritious and low-fat characteristics and has been cultivated for at least three millennia in China [1]. A dry Chinese chestnut kernel contains 50%–70% starch and is an alternative gluten-free flour source [2]. Currently, Chinese chestnut is widely grown in Asia, Europe, and America for nut production, accounting for more than 80% of the world's production, according to the most recent data collected by the Food and Agricultural Organization of the United Nations (2018, http://www.fao.org/faostat/en/#data/qc).

Starch consists of two major polysaccharides, namely amylose and amylopectin. Several studies on Chinese chestnuts have found that the starch content and ratio of amylose to amylopectin not only affect the quality and taste of the nuts but also are important factors in determining how the nuts are to be processed and their applications. Cultivars with a low amylose content are more suitable for the production of thermally processed chestnut kernels [3]. Yu et al. [4] reported a positive relationship between turbidity, gel hardness, and chewiness and the retrogradation degree of the starch to amylose content in Chinese chestnut, and negative relationships among swelling power, thermal enthalpy, and cohesiveness. The major component of Chinese chestnut starch is amylopectin, of which the content is approximately 2−3 times that of amylose in it [5]. Chinese chestnut flowers in mid-summer, and it takes approximately 100 days for nuts to fully ripen. Zhang et al. [5] reported that the dry mass of Chinese chestnut fruits increased 7.6-fold from 30 days after flowering (DAF) to 90 DAF, and the increase in dry mass in the fruit was primarily due to starch accumulation in the endosperm. While the contents of other metabolites, such as soluble sugars, proteins, and lipids, increased by weight, their ratios in the endosperm decreased until 75 DAF. Similar results were obtained by Chen et al. [6]. In another chestnut species, *C. henryi*, starch also accumulated rapidly and peaked at 109 DAF, while the sucrose level decreased [7]. After 109 DAF, the starch content slightly dropped and the sucrose level increased.

Studies of other plant species, particularly model species, have revealed that sucrose synthase (EC2.4.1.13) (SS) plays an important role in producing UDPG in sucrose and starch metabolism. According to the pathway depicted in the Kyoto Encyclopedia of Genes and Genomes (KEGG), UDPG produced by sucrose synthase is catalyzed by nucleotide diphosphatase (EC3.6.1.9) to form alpha-1-phosphate glucose, which reacts with ADP–glucose pyrophosphorylase (AGP, EC2.7.7.27) and ectonucleotide pyrophosphatase (EC3.6.1.9) to produce ADPG. Amylose is synthesized by starch granule synthase (EC2.4.1.242) and starch synthase (EC2.4.1.21) using ADPG as the substrate. Glycogen synthase (EC:2.4.1.11) and starch granule synthase can also use UDPG directly for the synthesis of amylose. Amylose is utilized to synthesize starch by the α-1,4-glucan branching enzyme (EC2.4.1.18). To date, studies on nut starch from chestnut have mainly focused on its physical and chemical properties, for example, the properties of its particles, hydrolytic properties, and gelatinization temperature, as well as its production and processing [4]. Limited information is available on starch and sucrose metabolism. Based on enzyme activity, Guo and Xie [8] discovered four types of synthase that may affect starch accumulation in the Chinese chestnut: ADP–glucose pyrophosphorylase, starch granule synthase (GBSS), soluble starch synthase (SSS), and amyloid branching enzyme (SBE).

Genomic tools, such as transcriptome analysis, have been utilized to understand the resistance of chestnut to blight [9], ink disease [10], and gall wasp [11]. Chen et al. [12] identified 18 Squamosa promoter-binding protein-like (SPL) transcription factor-encoding genes in the Chinese chestnut genome and revealed that miR156 cleaves SPL9 and SPL16. In terms of nut development, there are two reports currently available. Zhang et al. [5] compared the gene expression profiles of two stages (45 vs. 75 DAF) of Chinese chestnut and identified a total of 1537 differentially expressed unigenes. Since key cytosolic AGP-encoding genes (*brittle2* and *shrunken*) were not discovered in the transcriptome, it was suggested that the starch biosynthesis pathway of Chinese chestnut is similar to that of the potato tuber/Arabidopsis leaf and differs from that of maize endosperm. In a study by Chen et al. [6], two starch branching enzyme isoforms, CmSBE I and CmSBE II, were identified by zymogram analysis and found to reach their peak expression at 74 days after pollination (DAP) in the Chinese chestnut. Correspondingly, gene expression for these two CmSBE isoforms increased from 64 DAP and reached their highest levels at 77 DAP. These results correlated with the amylopectin levels during nut development, suggesting that the CmSBE enzymes contribute to amylopectin synthesis and influence the amylopectin content in the developing seed [6].

In order to enhance the transcriptomic resources and further understand starch and sucrose metabolism in maturing chestnuts, we included two new, later ripening stages (82 and 94 DAF), along with 70 DAF in a transcriptomic study of Chinese chestnut kernels. We hypothesized that starch

synthase and branching enzyme genes play important roles as chestnut kernel reaches maturity. These new insights provide valuable information for genetic improvement and breeding for nut quality in chestnut species.

## 2. Materials and Methods

### 2.1. Plant Materials

The materials used in the study were Chinese chestnut kernels from 15-year-old trees of "Yongfeng 1" (YF1, a fine southwestern cultivar in Yunnan Province, China) and "Yongren Zao" (YRZ, a fine southwestern cultivar in Yunnan Province, China). These trees were located in Weidi Township, Yongren County, Yunnan Province, at an altitude of 1600–1950 m. Starting approximately 20 days before maturation, kernels were collected every 11 days, and the last samples obtained were mature fruits with dehiscent burrs. The sampling dates were July 20, August 1, and August 13, 2017, corresponding to 70 DAF, 82 DAF, and 94 DAF (Figure 1). Each sample contained nuts from one burr, and three burrs were collected for each development stage. Fruit pericarps and seed coats were removed before kernel materials were either frozen in liquid nitrogen in the field and stored at −80 °C for RNA extraction or were dried for 12 h at 60 °C for analysis of sugar and starch contents.

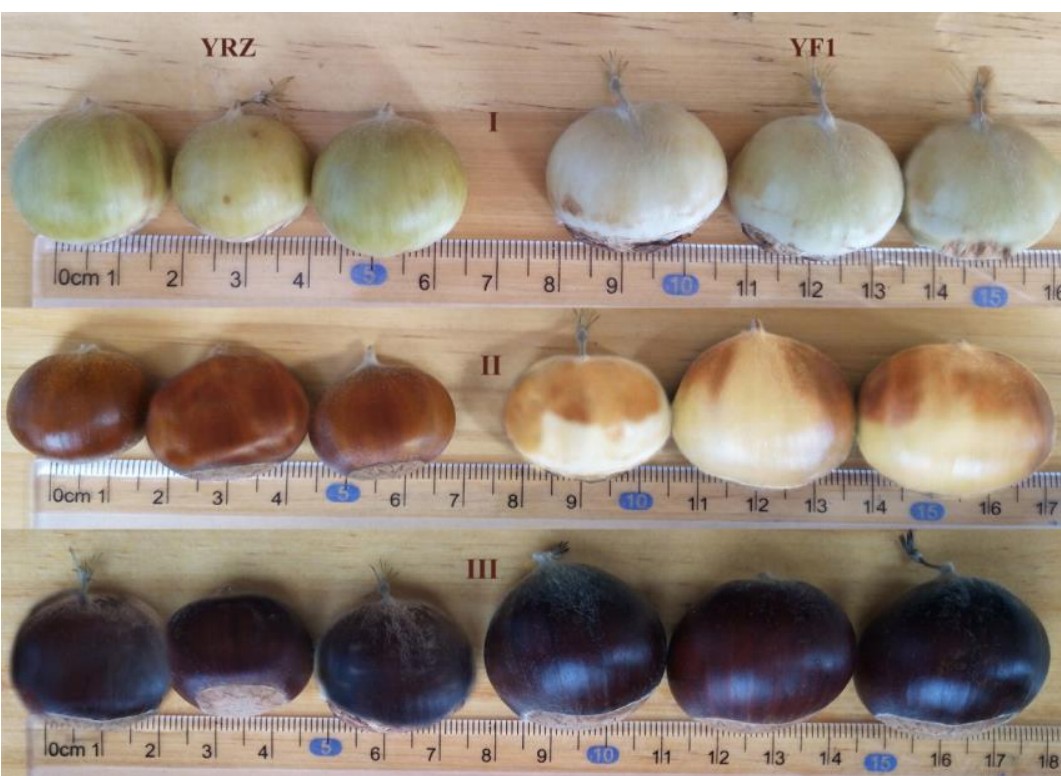

**Figure 1.** Illustration of seeds at three stages that were included in the study. I: 70 DAF, II: 82 DAF, III: 94 DAF. YF1, "Yongfeng 1" cultivar; YRZ, "Yongren Zao" cultivar.

### 2.2. Determination of Sugar and Starch Contents

Dried kernel materials were ground into a fine powder before being extracted with either double distilled water or 85% ethanol for sugar and starch analysis, respectively. The soluble sugar content was determined by anthrone colorimetry, as described in Oleksyn et al. [13]. To reduce the sugar content, a direct titration method was employed, in accordance with Alexander et al. [14]. Amylopectin and amylose contents were determined using a microanalytical method [15].

RNA Isolation and Library Construction

Total RNA from kernel samples at three time points of late nut development was isolated using RNAprep Pure Plant Kits (Tiangen Biotech, Beijing, China). The RNA concentration was determined using a spectrophotometer at A260/280 (NanoDrop 1000, Thermo Scientific, Wilmington, DE, USA), and the integrity was assessed using an Agilent 2100 system (Agilent Technologies). cDNA libraries were constructed individually for each sample with a VAHTS Stranded mRNA-seq Library Prep Kit. Paired-end sequencing was carried out on an Illumina 2500 (NR602, Vazyme, Nanjing, Jiangsu, China).

## 2.3. Assembly and Functional Annotation

Raw reads were trimmed and filtered to remove Illumina adapter sequences, low-quality reads, and reads containing poly-N sequences (>5%) by using in-house Perl scripts. The clean reads were then assembled de novo using Trinity software [16]. The unigenes were subjected to basic local alignment search tool searches [17] against the non-redundant protein sequence (NR), NT, Swiss-Prot, Kyoto Encyclopedia of Genes and Genomes (KEGG), COG, and Gene Ontology (GO) libraries. According to the NR annotation information, Blast2GO [18] was applied to obtain GO annotations of unigenes. After the GO annotation of each unigene, WEGO software [19] was used to classify and count the GO functions of all unigenes.

## 2.4. Differential Gene Expression Analysis

The unigenes from all 18 seed samples (two Chinese chestnut cultivars x 3 timepoints x 3 biological replicates) were clustered, and an all-unigene group was generated to serve as a reference transcriptome. The clean reads of each sample were then mapped back to the reference library using Bowtie2 software [20]. Normalized fragments per kilobase per million were used to quantify transcript abundance in the reads, and this information was used to compare the mRNA levels between samples. Cufflinks was used for the calculation of differential expression patterns. Genes with a q value ≤ 0.05 and fold change ≥ 1 were defined as differentially expressed genes (DEGs). Based on the GO functional classifications, GO terms with more than 100 DEGs were mapped. An enrichment analysis was conducted with KOBAS 2.0 [21], and the cutoff was FDR ≤ 0.05, which was adjusted by the Benjamini–Hochberg procedure (1995).

## 2.5. qRT-PCR Analysis

The expression patterns of 10 unigenes encoding SSS and SS in developing Chinese chestnut seeds were studied by qRT-PCR. Total RNA was extracted using a Qiagen RNeasy Mini Kit (Qiagen Inc., Valencia, CA) and was reverse transcribed into cDNA by random primers. The names and primer sequences of the 10 unigenes are listed in Table S1. The internal control was 18S rRNA. The $2^{(-\triangle\triangle CT)}$ method was employed to analyze the data [22].

## 3. Results

### 3.1. Dynamics of Sugar and Starch Contents at Different Ripening Stages

A comparison of the sugar content at different periods revealed that both soluble and reducing sugar levels reached the highest values at stage I (70 DAF) and the lowest values at stage II (82 DAF) (Table 1). Starch analysis showed that the amylopectin content gradually increased during nut maturation, with the smallest amount detected at stage I and the highest amount observed at stage III. The dynamics of the total starch content were the same as those of the amylopectin content, whereas the amylose content decreased by stage III. Similar to the amylopectin and total starch contents, the amylopectin/amylose ratio continued to increase as the nuts matured. Sugar and starch contents showed opposite trends at stages I and II. Stage I nuts showed peak sugar accumulation, and mature chestnuts (stage III) mainly exhibited starch accumulation. While both cultivars showed similar sugar

and starch content dynamics at different ripening stages, the Yongfeng 1 cultivar had higher contents of amylopectin and total starch, and Yongren Zao contained more total sugar by 94 DAF.

**Table 1.** Analysis of significant differences in sugar and starch contents.

| | Soluble Sugar (%) | Reducing Sugar (%) | Total Sugar (%) | Amylopectin (mg/g) | Amylose (mg/g) | Total Starch (mg/g) | Amylose/Amylopectin Ratio (%) |
|---|---|---|---|---|---|---|---|
| I-YF1 | 19.27 ± 0.10b | 2.57 ± 0.90a | 21.84 ± 0.19b | 283.70 ± 2.45f | 133.12 ± 1.69a | 516.83 ± 1.77d | 46.92a |
| II-YF1 | 7.34 ± 0.06d | 1.57 ± 0.10d | 8.90 ± 0.42f | 445.36 ± 2.95c | 143.05 ± 1.75a | 588.41 ± 2.36b | 32.12c |
| III-YF1 | 15.54 ± 0.41c | 1.81 ± 0.02c | 17.35 ± 0.41d | 519.68 ± 4.66a | 102.01 ± 1.00b | 621.69 ± 5.67a | 19.62d |
| I-YRZ | 26.25 ± 0.56a | 2.15 ± 0.11b | 28.40 ± 0.10a | 237.00 ± 0.05e | 90.60 ± 0.24c | 327.60 ± 2.27e | 38.22b |
| II-YRZ | 9.25 ± 0.10d | 1.51 ± 0.51d | 10.76 ± 0.12e | 403.33 ± 1.87d | 146.24 ± 0.62a | 549.57 ± 2.46c | 36.26b |
| III-YRZ | 17.17 ± 0.55bc | 1.77 ± 0.09c | 18.9 ± 0.61c | 484.80 ± 0.1bc | 106.56 ± 0.48b | 591.36 ± 0.56b | 21.98d |

Note: Lower case letters indicate significant differences ($P < 0.05$); I, II and III represent the three development stages (July 20, August 1, and August 13, 2017, respectively).

### 3.2. Sequencing Data and DEG Statistics

The transcriptome sequencing analysis generated at least 44.82 Mb of raw data for each sample, and after sequencing quality control, at least 43.51 Mb of clean data was obtained for each sample. The percentage of Q30 bases in each sample was ≥92.92%. At least 56,725 unigenes were obtained for each sample, and the value of N50 was greater than 1403 bp (Table S2). Sequencing data are available via the NCBI Bioproject PRJNA574282.

The transcriptome data generated a total of 181,998 unigenes, among which 126,647 had an annotation, accounting for 69.59% of all unigenes. In particular, annotations of 39,506 unigenes overlapped among the NR, NT, Swiss-Prot, KEGG, and COG libraries (Figure 2A). Based on the NR annotations and the E-value distribution, 82.9.1% and 60.6% of the mapped sequences showed strong homology (E-value <10−15) and very strong homology (E-value <10−45), respectively, to the available plant sequences (Figure 2B). The eight top-hit species based on the Nr annotations are shown in Figure 2C. Nearly 64% of the unigenes could be annotated with sequences from the top three species: *Vitis vinifera*, *Amygdalus persica*, and *Ricinus communis* (Figure 2C).

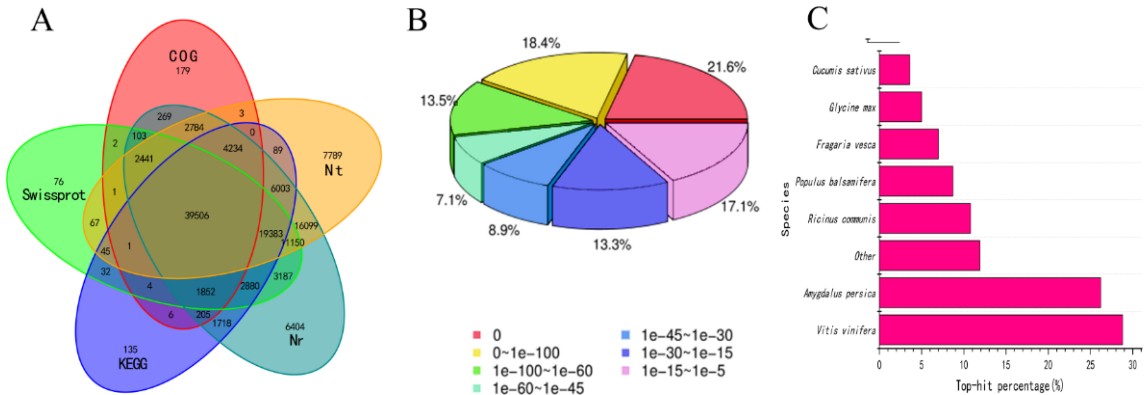

**Figure 2.** Characteristics of the homology search for Chinese chestnut kernel unigenes. (**A**) Venn diagram of the number of unigenes annotated by BLASTx with an E-value threshold of $10^{-5}$ against protein databases. The numbers in the circles indicate the numbers of unigenes annotated by single or multiple databases. (**B**) E-value distribution of the top BLASTx hits against the NR database for each unigene. (**C**) Number and percentage of unigenes matching the top eight species using BLASTx in the NR database.

When comparing the DEGs of the three developmental stages in pairs, it was found that 5664 DEGs were upregulated and 4266 were downregulated in I-vs-II, 5520 genes were upregulated and 3489 were downregulated in II-vs-III, and 13,734 were upregulated and 9283 were downregulated in I-vs-III (Figure 3). The numbers of DEGs were similar between the I-vs-II and II-vs-III comparisons.

In contrast, the number of DEGs was the largest in the I-vs-III comparison, indicating a significant difference between these two stages.

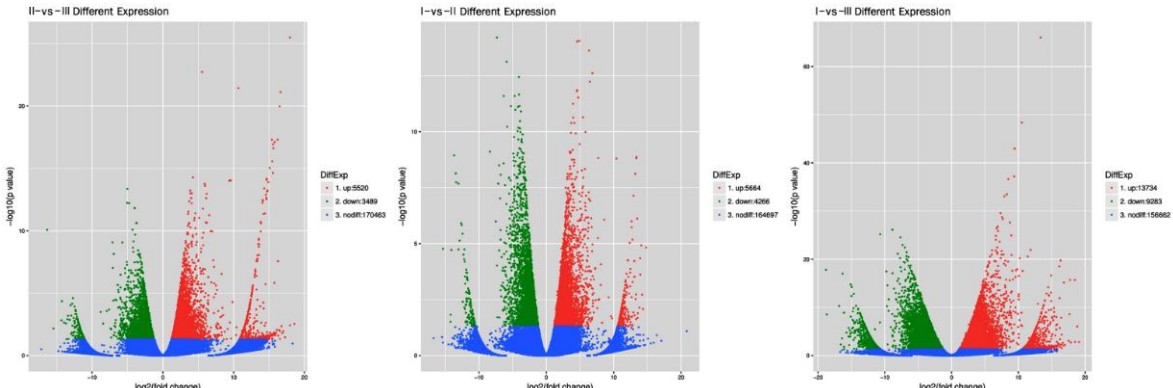

**Figure 3.** Volcano map of differentially expressed unigenes in the II-VS-III, I-VS-II, and I-VS-III comparisons. I, II, and III represent the three sampling dates: 70 DAF, 82 DAF, and 94 DAF.

*3.3. GO Annotation Analysis of DEGs*

An analysis of all DEGs revealed that the cellular components contained the highest number of genes, followed by molecular function. In the biological process, the DEGs identified in the I-vs-II comparison were related to the cell wall, whereas the differences between stage I and stage III were associated with various ion transport pathways. For the cellular components, GO terms identified in the II-vs-III comparison were also annotated in the I-vs-III comparison, with the exception of plant-type cell wall. GO terms in the I-vs-II comparison overlapped with the ones identified in the I-vs-III and II-vs-III comparisons, with the exceptions of apoplasts and microtubules. The I-vs-III comparison had a distinct annotation in vacuoles that was not found in the other two comparisons. As for the molecular function, the GO terms in the I-vs-II comparison included hydrolase and oxidoreductase activities, the II-vs-III comparison had sodium:proton antiporter, sodium ion transmembrane transporter, and antiporter activities, and the I-vs-III comparison had all activities identified in the II-vs-III comparison plus the monovalent cation:proton antiporter, solute:proton antiporter, cation:cation antiporter, and oxidoreductase. Overall, the GO terms obtained in the I-vs-II and II-vs-III comparisons differed the most, sharing four cell component categories (Figure 4).

*3.4. KEGG Pathway Analysis of DEGs*

Many KEGG pathways were identified in the DEGs. Like in other plant species, the biosynthesis of secondary metabolites and metabolic pathways contained the most DEG genes, followed by the 10 pathways shown in Figure 5 for each comparison. The common DEG-enriched pathways among the three comparisons were plant hormone signal transduction, starch and sucrose metabolism, plant–pathogen interactions, purine metabolism, RNA degradation, pyridine metabolism, glycerophospholipid metabolism, and RNA transport (Figure 5). In some metabolic pathways, the DEGs showed differences among different pairs of comparison. The metabolic pathways of ribosome biogenesis in eukaryotes and RNA polymerase were identified in the I-vs-II comparison but were absent in the II-vs-III and I-vs-III comparisons. In contrast, endocytosis and spliceosome metabolic pathways were found in the II-vs-III and I-vs-III comparisons but were absent in the I-vs-II comparison. We can thus infer that as chestnut kernels ripen, the main metabolic pathways in the chestnut seed kernels change from ribosome biogenesis in eukaryotes (KEGG pathway: 03008) and RNA polymerase (KEGG pathway: 03020) to endocytosis (KEGG pathway: 04144) and spliceosomes (KEGG pathway: 03040).

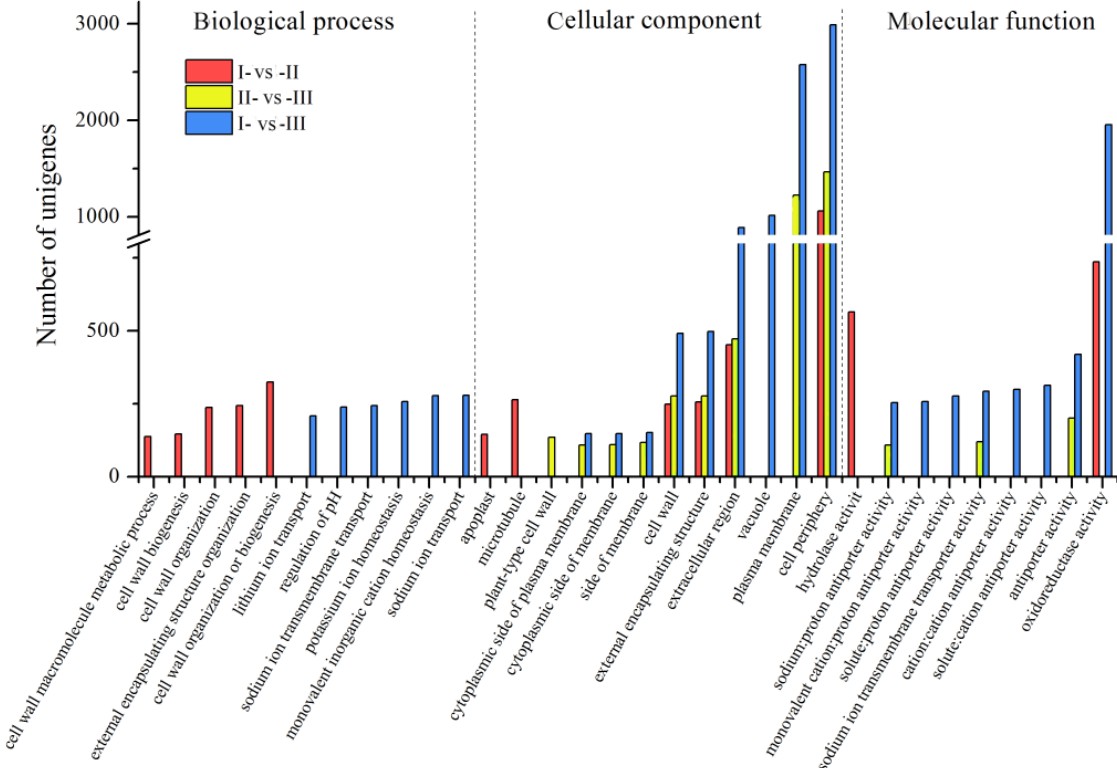

**Figure 4.** Comparison of Gene Ontology (GO) annotation categories of the Chinese chestnut kernel at three developmental stages.

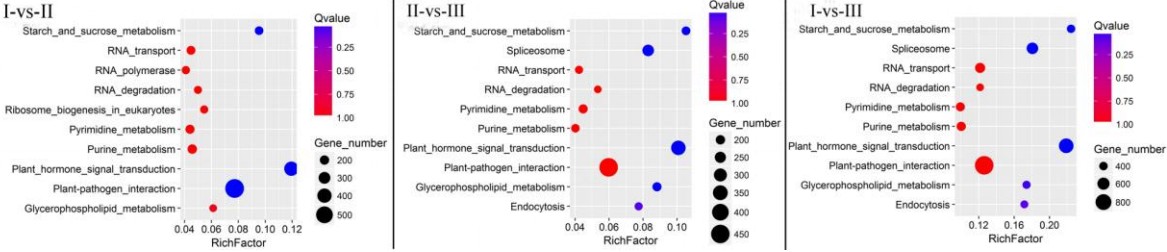

**Figure 5.** Distribution of Chinese chestnut kernel differentially expression genes (DEGs) in Kyoto Encyclopedia of Genes and Genomes (KEGG) pathways. (Note: The abscissa shows the enrichment factor, and the ordinate shows the KEGG pathway).

*3.5. DEG Analysis of Sucrose and Starch Metabolic Pathways*

To better understand the dynamics of the sucrose-to-starch metabolic pathways during the ripening of Chinese chestnut, key sections of the pathways were examined. In the I-vs-II comparison, all four identified sucrose synthase DEGs were downregulated, and starch synthase had no DEGs. In the II-vs-III comparison, eight sucrose synthase DEGs were upregulated and seven were downregulated, while the sole starch synthase DEG was upregulated. In the I-vs-III comparison, 24 sucrose synthase DEGs were upregulated and 25 were downregulated, while all 24 starch synthase DEGs were upregulated (Figure 6, Tables 2 and 3).

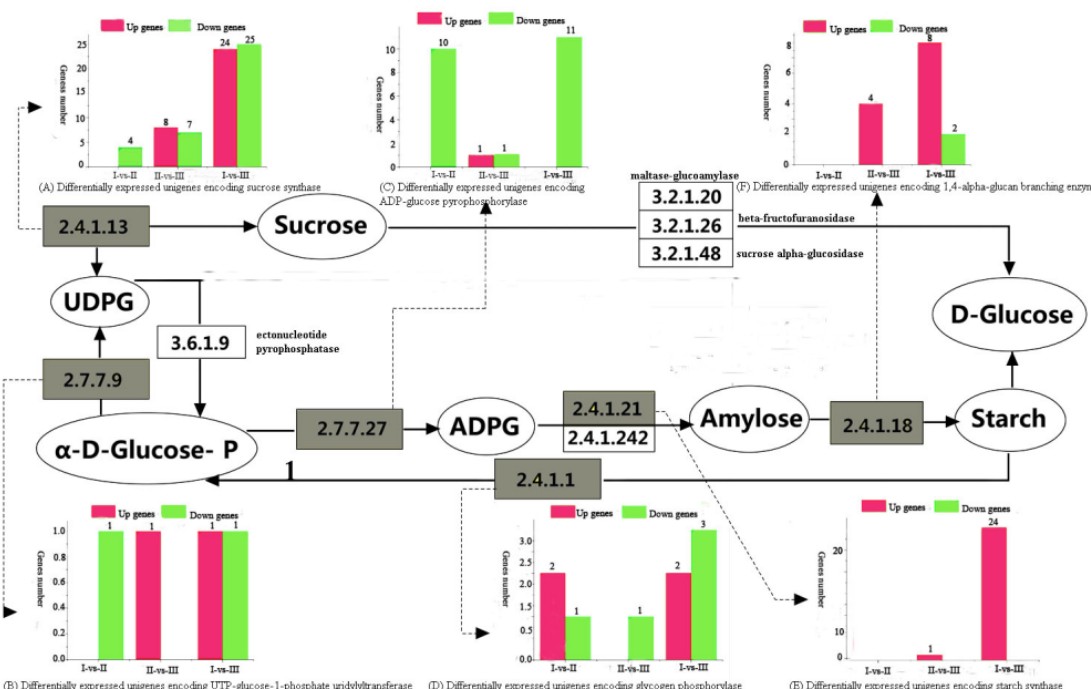

**Figure 6.** Major KEGG pathways for starch and sucrose metabolism in Chinese chestnut kernel development. Major enzymes with differential gene expression are highlighted. The numbers of upregulated and downregulated genes during the three developmental stages are shown: (**A**) sucrose synthase, (**B**) UTP–glucose-1-phosphate uridylyltransferase, (**C**) ADP–glucose pyrophosphorylase, (**D**) glycogen phosphorylase, (**E**) starch synthase, and (**F**) 1,4-alpha-glucan branching enzyme.

As shown in Figure 6, All ADP–glucose pyrophosphorylase (AGPase) DEGs were downregulated in the I-vs-II and I-vs-III comparisons, while both up- and down- regulated AGPase genes were found in the II-vs-III comparison. The α-1,4-glucan branching enzyme had no DEGs in the I-vs-II comparison, whereas all DEGs were upregulated in the II-vs-III comparison, and both upregulation and downregulation were observed in the I-vs-III comparison. UDPG is produced by the reaction of α-1-phosphate glucose with UDP-glucose-1-phosphate esterase transferase (EC2.7.7.9). UDP-glucose-1-phosphoesterase DEGs were all downregulated in the I-vs-II comparison, whereas the opposite was observed in the II-vs-III comparison. As for the I-vs-III comparison, both up- and down-regulated DEGs were found. Starch is decomposed into α-1-glucose phosphate by glycogen phosphorylase (EC2.4.1.1). As shown in Figure 6, the DEGs encoding this enzyme were downregulated in the II-vs-III comparison, which suggests that the activity of glycogen phosphorylase is relatively low in stage III.

Based on the kernel coloration, the Yongren Zao cultivar seems to mature earlier than Yongfeng 1 (Figure 1). Correspondingly, the RNAseq data show that a sucrose synthetase unigene (Unigene49637_All) has lower expression in Yongren Zao than Yongfeng 1 at all three stages (Table 4). There were other sucrose metabolism-related unigenes that showed different expression levels between the two cultivars: sucrose–phosphate synthase, fructokinase, trehalose 6-phosphate synthase, trehalose 6-phosphate phosphatase, beta-glucosidase, and UTP–glucose-1-phosphate uridylyltransferase. However, none of the uninegenes encoding starch synthesis enzymes were differentially expressed between Yongren Zao and Yongfeng 1.

**Table 2.** List of differentially expressed sucrose synthase [2.4.1.13] genes.

| Unigene (I-VS-II) | Unigene (II-VS-III) | Unigene (I-VS-III) |
| --- | --- | --- |
| Unigene66368_All, | Unigene40270_All, | Unigene31283_All, |
| Unigene51456_All, | Unigene31283_All, | Unigene40270_All, |
| Unigene62017_All, | Unigene75288_All, | Unigene75288_All, |
| Unigene67714_All, | CL12365.Contig42_All, | CL12365.Contig42_All, |
| | CL12365.Contig38_All, | CL12365.Contig38_All, |
| | Unigene7815_All, | CL12365.Contig36_All, |
| | CL12365.Contig34_All, | Unigene7815_All, |
| | CL11646.Contig2_All, | CL12365.Contig37_All, |
| | Unigene12460_All, | CL12365.Contig19_All, |
| | Unigene30937_All, | CL12365.Contig33_All, |
| | Unigene30947_All, | CL12365.Contig12_All, |
| | Unigene30932_All, | CL12365.Contig3_All, |
| | Unigene80971_All, | CL12365.Contig21_All, |
| | Unigene30942_All, | CL12365.Contig25_All, |
| | Unigene89949_All, | CL12365.Contig23_All), |
| | | CL12365.Contig24_All, |
| | | CL12365.Contig14_All, |
| | | CL12365.Contig41_All, |
| | | CL12365.Contig4_All, |
| | | CL12365.Contig34_All, |
| | | CL12365.Contig5_All, |
| | | CL12365.Contig43_All, |
| | | CL11646.Contig2_All, |
| | | CL12365.Contig9_All, |
| | | Unigene51456_All, |
| | | Unigene67714_All, |
| | | Unigene62017_All, |
| | | Unigene66368_All, |
| | | Unigene12460_All, |
| | | Unigene30937_All, |
| | | Unigene30947_All, |
| | | Unigene80971_All, |
| | | Unigene30932_All, |
| | | Unigene30942_All, |
| | | Unigene30941_All, |
| | | Unigene89949_All, |
| | | Unigene30934_All, |
| | | Unigene30945_All, |
| | | Unigene27481_All, |
| | | Unigene30946_All, |
| | | Unigene80976_All, |
| | | Unigene80977_All, |
| | | Unigene30944_All, |
| | | Unigene57545_All, |
| | | Unigene80978_All, |
| | | Unigene84850_All, |
| | | Unigene80975_All, |

I, II, and III represent the three sampling dates: July 20, August 1, and August 13, 2017. LogFC was at least 2. The red letters indicate upregulated genes, and the green letters indicate downregulated genes.

**Table 3.** List of differentially expressed starch synthase [EC2.4.1.21] genes.

| Unigene (I-VS-II) | Unigene (II-VS-III) | Unigene (I-VS-III) |
|---|---|---|
| —— | CL2750.Contig1_All | CL12336.Contig8_All, |
| | | CL12336.Contig40_All, |
| | | CL12336.Contig10_All, |
| | | CL12336.Contig46_All, |
| | | CL12336.Contig26_All, |
| | | CL12336.Contig1_All, |
| | | CL12336.Contig3_All, |
| | | CL12336.Contig13_All, |
| | | CL12336.Contig4_All, |
| | | CL12336.Contig14_All, |
| | | CL12336.Contig2_All, |
| | | CL12336.Contig15_All, |
| | | CL12336.Contig22_All, |
| | | CL12336.Contig25_All, |
| | | CL12336.Contig5_All, |
| | | CL12336.Contig41_All, |
| | | CL12336.Contig31_All, |
| | | CL12336.Contig28_All, |
| | | CL12336.Contig11_All, |
| | | CL12336.Contig30_All, |
| | | CL12336.Contig44_All, |
| | | CL2750.Contig2_All, |
| | | CL2750.Contig11_All, |
| | | CL2750.Contig1_All |

I, II, and III represent the three sampling dates: July 20, August 01, and August 13, 2017. LogFC was at least 2. The red letters indicate upregulated genes.

**Table 4.** The sucrose synthase and starch synthase genes of Yongfeng 1 (YF1) and Yongren Zao (YRZ) in the sucrose and starch metabolic pathways during the three stages.

| YF1-vs-YRZ | Annotation | Genes |
|---|---|---|
| | Sucrose synthetase[2.4.1.13] | Unigene23727_All |
| | Sucrose-phosphate synthase[2.4.1.14] | Unigene23727_All |
| I | Fructokinase [2.7.1.4] | CL11976.Contig1_All |
| | Trehalose 6-phosphate synthase[2.4.1.15] | Unigene12426_All |
| | Trehalose 6-phosphate phosphatase[3.1.3.12] | Unigene12426_All |
| | Sucrose synthetase [2.4.1.13] | Unigene23727_All |
| | Sucrose-phosphate synthase [2.4.1.14] | Unigene23727_All |
| II | beta-glucosidase [3.2.1.21] | CL7021.Contig36_All |
| | Trehalose 6-phosphate synthase [2.4.1.15] | Unigene12426_All |
| | Trehalose 6-phosphate phosphatase [3.1.3.12] | Unigene12426_All |
| | Sucrose synthetase [2.4.1.13] | CL6258.Contig2_All, Unigene49637_All, CL11646.Contig2_All |
| | beta-glucosidase [3.2.1.21] | CL7021.Contig3_All |
| III | UTP–glucose-1-phosphate uridylyltransferase[2.7.7.9] | Unigene95924_All |
| | beta-amylase [3.2.1.2] | Unigene54154_All |
| | Fructokinase [2.7.1.4] | CL2613.Contig2_All |
| | Trehalose 6-phosphate synthase [2.4.1.15] Trehalose 6-phosphate phosphatase [3.1.3.12] | Unigene12426_All Unigene12426_All |

I, II, and III represent the three sampling dates. LogFC was at least 2. The red letters indicate upregulated genes and the green letters indicate downregulated genes.

*3.6. Verification by qRT-PCR*

To verify the reliability of the RNA sequencing data, qPCR was used to evaluate the expression profiles of seven up-regulated and three down-regulated genes encoding SS and SSS genes. The expression trends of the ten selected DEGs were consistent with the transcriptome data (Figure 7), indicating that both methods are reliable and complementary for estimating gene expression.

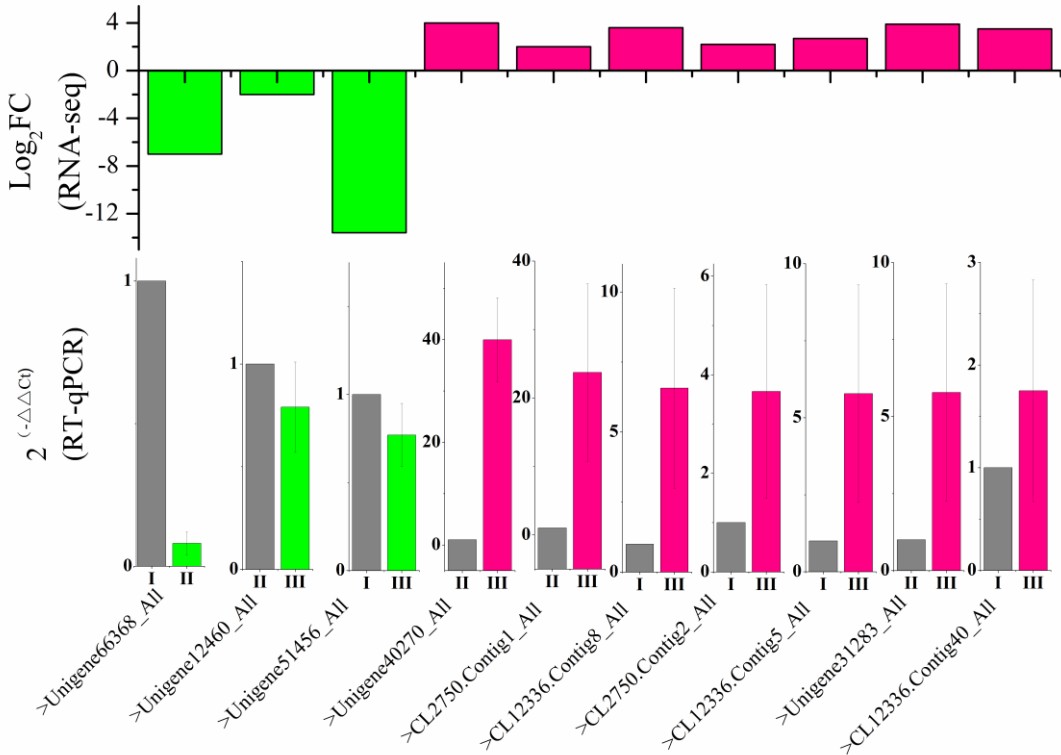

**Figure 7.** Expression of ten selected genes measured by RNA sequencing (upper panel) and qRT-PCR (lower panel). Error bars represent standard deviations.

## 4. Discussion

Chestnut is an important and popular food in many countries due to its nutritional compounds such as carbohydrates, polyphenols, vitamins, iron, folate, dietary fibers, minerals, and unsaturated fatty acids. As the most frequently produced chestnut in the world, fresh Chinese chestnut fruits contain 52.0% water, 42.2% carbohydrates, 4.2% proteins, and 0.7% lipids [23]. Starch is the main carbohydrate, followed by sucrose [24]. These two compounds are closely related to the physicochemical properties and quality of chestnut fruit. Data from the current study indicate that amylopectin and total starch continue to accumulate as nuts mature (from 70 to 94 DAF) in both cultivars—Yongfeng 1 and Yongren Zao. In contrast, the percentages of soluble and reducing sugars decreased in the late Chinese chestnut maturation stage. Zhang et al. [5] and Chen et al. [6] reported the same trend from 45 to 75 DAF and 60 to 80 DAP in Chinese chestnut, respectively. A similar dynamic pattern was found in wheat with the soluble sugar content decreasing as the grain starch content increased [25]. According to Wang et al. [26], transformation of sucrose in grains is an important factor for starch accumulation. After all, the products of sucrose cleavage are used in many metabolic pathways, including the synthesis of complex carbohydrates such as starch [27] (Figure 6).

The metabolism of starch and sucrose during chestnut development is closely related to the activity of corresponding enzymes, including sucrose synthase, starch synthase, UDP-glucose-1-phosphatase, α-1,4-glucan branching enzyme, and ADP–glucose pyrophosphorylase. Compared with the 70 and 82 DAF stages, starch synthase unigenes were significantly upregulated at 94 DAF. All of these up-regulated unigenes encode for soluble starch synthase. Another major type of starch synthase,

granule-bound starch synthase, was not found to have differential gene expression. In comparison, two granule-bound starch synthase unigenes had two-fold higher expression at 75 DAF than at 45 DAF [5]. This type of starch synthase functions mainly to synthesize amylose, as evidenced by a reduced amylose content or a complete lack of amylose in the lost-of-function mutants of rice, maize, and wheat [28]. Furthermore, all DE $\alpha$-1,4-glucan branching enzyme-encoding unigenes in II-vs-III were upregulated, and eight out of ten were upregulated in I-vs-III. This enzyme converts amylose into amylopectin by attaching short glucosyl chains with $\alpha$-1,6-glucosyl bonds to glycogen. Therefore, the active gene expression of soluble starch synthase and branching enzyme as well as the lack of change in granule-bound starch synthase gene activity, are the underlining mechanisms for the increase in the amylopectin/amylose ratio in maturing chestnuts. As for sucrose synthase, only down-regulated DE unigenes were found in I-vs-II, while almost equal numbers of up- and down-regulated DE sucrose synthase unigenes existed in I-vs-III and II-vs-III. This can be attributed to the fact that the reaction catalyzed by sucrose synthase is reversible and the products of sucrose cleavage are utilized for many other metabolic pathways in addition to the synthesis of complex carbohydrates [27]. Similarly, the activity of starch degrading enzymes was found to be low. For example, down-regulated DE unigenes encoding for glycogen phosphorylase, which catalyzes the phosphorolysis of glycogen to produce alpha-D-glucose-1-phosphate, were found in Stages II and III. Other starch degradation enzymes, such as amylase, phosphoglucan phosphatase DSP4, and maltose exporter did not show differential gene expression. It is noteworthy that DE unigenes encoding ADP-glucose pyrophoshorylase, an enzyme catalyzing the production of the plant's major glucosyl donor for starch synthesis, ADPG, were all down-regulated in stages II and III. This suggests that ADPG was no longer being actively produced by 82 DAF. Instead, the accumulated ADPG was being used by starch synthase for starch production.

In the qRT-PCR data presented by Zhang et al. [5] for Chinese chestnut, all five sucrose synthase-encoding unigenes had an overall decreased expression from 30 to 90 DAF with the lowest level at 75 DAF. Starch synthase-encoding unigenes exhibited the opposite trend, with two unigenes having the highest expression at 75 and 90 DAF and one peaking at 75 DAF. Our results corroborate the data presented by Zhang et al. [5]. Similarly, in rice, the starch content was found to be positively correlated with the activity of starch synthase [29]. Our results also suggest that UDP–glucose-1-phosphate uridylyltransferase is more active in stages I and III, with lower activity in stage II (Figure 6).

GO annotation and KEGG pathway analyses suggest that various metabolic relationships to come into play during the ripening of the chestnut kernel. In addition to starch and sucrose metabolism, RNA transport, RNA degradation, pyrimidine metabolism, purine metabolism, plant hormone signal transduction, plant–pathogen interactions, and glycerophospholipid metabolism were significantly enriched in all of the comparisons included in the study (Figure 5). It is intriguing that plant–pathogen interactions were enriched. As Chinese chestnut matured, the unique enriched pathways switched from ribosomal biogenesis and RNA polymerase in eukaryotes to endocytosis and spliceosomes. In order to fully understand starch biosynthesis and its regulation, it is important to investigate these enriched pathways and associated genes in the future.

ADPG is mainly synthesized in the cytosol by cytosolic AGP, which is encoded by *brittle2* and *shrunken2* genes in cereal endosperms such as maize [30,31]. ADPG is transported into amyloplasts by the brittle1 ADPG transporter [31]. In other plant tissues, such as potato tubers, Arabidopsis leaves, and pea embryos, ADPG is mainly synthesized in the plastids by plastidic AGP, and ANT is mainly used to exchange ATP/ADP across plastid envelopes [32]. Because no *brittle2* and *shrunken* were identified at the 45 and 75 DAF Chinese chestnut transcriptomes except for "brittle1", Zhang et al. [5] suggested that Chinese chestnut employs a similar biochemical mechanism to that used by Arabidopsis leaves and potato tubers. None of these genes were differentially expressed in our study of chestnuts at late maturing stages of 70, 82, and 94 DAF. It is possible that the relocation of ADPG is weak, if not diminished, for starch synthesis by late nut maturation. As discussed above, our data suggest that ADPG is no longer being actively produced by 82 DAF.

## 5. Conclusions

This is the first study to employ transcriptome sequencing for gene discovery and the study of gene functions in the late chestnut maturation stage. The new genomic resources and insights lay the foundation for genetic improvement and breeding for nut quality in chestnut species. Comparative analyses suggest that the active gene expression of soluble starch synthase and branching enzyme as well as the lack of change in granule-bound starch synthase gene activity are the underlining mechanisms for the increases in starch content and the amylopectin/amylose ratio in ripening chestnut, validating our hypothesis on the roles of starch synthase and branching enzyme genes.

**Supplementary Materials:** The following are available online at http://www.mdpi.com/1999-4907/11/4/388/s1, Table S1: Unigenes and sequences of 10 primer pairs for qPCR analysis, Table S2: Statistical analysis of transcriptome sequencing of *Castanea mollissima* BL.

**Author Contributions:** S.L. designed and performed the experiments, analyzed the data, drafted the paper, and prepared figures and tables. Z.S. and Z.Z. conceived the study and acquired funding for the project. H.L. gave advice on the project and organized, drafted, and revised the manuscript. L.T., L.X. and W.L. assisted with the experiments. All authors have read and agreed to the published version of the manuscript.

**Funding:** This work was supported by a grant from the Science and Technology Major Project of Guangxi (Guike AA17204058-03) and the Independent project of Key Laboratory of Cultivation and Utilization of Guangxi Characteristic Economic Forest(16-A-0401).

**Acknowledgments:** The authors thank Anpei Zhou for him critical reading of the manuscript.

**Conflicts of Interest:** The authors declare no conflict of interest.

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
