# Peer review of "Transcriptome Sequencing and Differential Expression Analysis Reveal Molecular Mechanisms for Starch Accumulation in Chestnut"

_forests, doi:10.3390/f11040388_

Round 1

Reviewer 1 Report

The authors have performed transcriptome analysis of Castanea mollissima in regard to starch and sucrose metabolism. The results have confirmed a similar metabolic pathway as has been found in other/model species. The results are especially interesting in the light of the economic importance of the species.

I have little criticism towards the presentation of the paper:

  1. The introduction should be better focused on the goal and novelty of the study; try to formulate a hypothesis that you have tried to test.
  2. The conclusions are written as a kind of abstract. It would be better to put the last paragraph of the discussion into the conclusions and work out more, what you really can conclude by the data that you have found and their importance to understand the physiologic pathways in the starch development.
  3. Have a native speaker to perform a final language check; e.g. Chinese chestnuts are very important producers of chestnuts, but are not by themselves the "largest" producer (see line 16 and 278f).

Reviewer 2 Report

The present manuscript expands the transcriptomic data by Zhang et al. (2015. J. Agric. Food Chem., 63, 3, 929-942) by analyzing the transcriptomes in later ripening stages of kernels from two Chinese chestnut cultivars. One of the main concerns about the presented data, however, relates to the extent of the novelty. As the presented data mostly corroborates previous knowledge, the novel findings must be better summarized and (structurally) differentiated from those corroborating previous results in the Conclusions, for clarity to readers unfamiliar with the topic.

While the sugar and starch content determination are performed in two different cultivars over 3 ripening stages, the transcriptome analysis focuses on the maturation stages but excludes comparative analysis between cultivars. As kernels from YF1 and YRZ show phenotypic differences (Fig1), as well as differences in sugar and starch content at a given stage (Table1), a transcriptomic comparative of YF1 and YRZ cultivars along the stages, at least for sucrose and starch metabolic pathways, should be provided and discussed.

Other issues:

  • Abstract: there is an enormous jump from 1st and 2nd sentence to the 3rd This affects the reading and context comprehension considerably.
  • Along the text, please use “at” or no preposition before an acronym such DAF.
  • Line 108, “being for being” sounds like a typo
  • Line 197, Fig3 does not illustrate the text
  • Line 217, Fig4 does not illustrate the text
  • Line 269, it is stated that “The expression trends of the ten selected DEGs were consistent with the transcriptome data”, but according to fig7 this is not really the case for 2 of the genes.
  • Figure3 needs a more descriptive figure legend.
  • Line 284_ the data in reference 6 comes from cotyledons. Why the authors assume that developmental dynamics in starch and sugar contents in cotyledon and kernel are comparable must be argued and referenced if appropriate.
  • 94 DAF and 95 DAF are sometimes indistinctly used in the text. Check and standardize.
  • Could the authors discuss why different sucrose synthases are upregulated while others are downregulated over the ripening process, and how that might correlate with the regulation of starch contents in chestnuts?
  • Line 307, please avoid mixing the concepts of enzyme activity and transcriptional response as in the sentence “activity of starch degrading enzymes seems low”
  • Line 341, weak?

Round 2

Reviewer 2 Report

I thank the authors for addressing my comments. I wish them all the best.